# Suicidal ideation and associated factors among pregnant women attending antenatal care in Jimma medical center, Ethiopia

Tamrat Anbesaw[1]*, Alemayehu Negash[2], Almaz Mamaru[3], Habtamu Abebe[4], Asmare Belete[1], Getinet Ayano[5,6]

1 Department of Psychiatry, College of Medicine and Health Science, Wollo University, Dessie, Ethiopia, 2 Department of Psychiatry, Institute of Health Sciences, Faculty of Medical Science, Jimma University, Jimma, Ethiopia, 3 Department of Psychiatry, Faculty of Medical Science, Institute of Health Sciences, Jimma University, Jimma, Ethiopia, 4 Department of Epidemiology and Biostatistics, Faculty of Medical Science, Institute of Health Sciences, Jimma University, Jimma, Ethiopia, 5 Research and Training Department, Amanuel Mental Specialized Hospital, Addis Ababa, Ethiopia, 6 School of Public Health, Curtin University, Bentley, Australia

* tamratanbesaw@gmail.com

**Data Availability Statement:** All relevant data are within the manuscript and its Supporting Information files.

## Abstract

### Background

Suicidal ideation (SI) among pregnant women is a major public health concern worldwide and is associated with a higher risk of completed suicide. However, there are limited studies that determined the prevalence and the potential determinants of suicidal ideation in Sub-Saharan Africa, including Ethiopia. Therefore, this study aimed to explore the prevalence of suicidal ideation and associated factors among pregnant women attending antenatal care in Jimma, Ethiopia.

### Methods

An institutional-based cross-sectional study was conducted among 423 pregnant women attending Jimma medical center in Southwest, Ethiopia. A systematic random sampling technique was used to select the study participants. Suicidal ideation assessed using the Suicidality Module of the World Mental Health survey initiative version of the World Health Organization Composite International Diagnostic Interview (CIDI). Other tools used are EPDS, Abuse Assessment Scale (AAS), DASS -21, PSS, Maternity Social Support Scale (MSSS), and Pittsburgh Sleep Quality Index (PSQI). A multivariable logistic regression analysis was used to explore the potential determinants of suicidal ideation among the participants.

### Result

The prevalence of SI among women who are on antenatal care was found to be 13.3% (95% CI (10.1,16.4). In multivariable analysis, marital status with lack of cohabiting partners (AOR = 2.80,95%CI:1.23,6.37), history of abortion (AOR = 2.45,95% CI:1.03,5.93), having depression (AOR = 4.28,95% CI:1.75,10.44), anxiety(AOR = 2.99,95% CI:1.24,7.20), poor

**Funding:** This study was funded by Jimma University. The funders had no role in study design, data collection, and analysis, decision to publish, or preparation of the manuscript.

**Competing interests:** The authors have declared that no competing interests exist.

**Abbreviations:** AAS, Abuse Assessment Screen; AIDS, Acquired Immune Deficiency Syndrome; AKUADS, Aga Khan University Anxiety and Depression Scale; ANC, Antenatal Care; CI, Confidence Interval; CIDI, Composite International Diagnostic Interview; EPDS, Edinburgh Postnatal Depression Scale; Epi-Data, Epidemiological Data; IPV, Intimate Partner Violence; JMC, Jimma Medical Center; MDD, Major Depressive Disorder; MINI, Mini-International Neuropsychiatric Interview; OR, Odds Ratio; SI, Suicidal Ideation; SPSS, Statistical Package for Social Science; US, United States; WHO, World Health Organization.

sleep quality (AOR = 2.85,95% CI:1.19,6.79), stress (AOR = 2.50, 95% CI:1.01,5.67), and intimate partner violence (AOR = 2.43, 95% CI:1.07,5.47) were found to be significant predictors of suicidal ideation.

## Conclusion

The prevalence of SI among pregnant women was found to be huge. Lack of cohabiting partners, previous history of abortion, depression, anxiety, intimate partner violence, poor sleep quality, and stress were variables that are independent predictors of suicidal ideation. Screening and interventions of antenatal SI are needed.

## Background

Suicide is a fatal act of terminating one's own life [1]. Suicidal ideation (SI) is a thought about one's serving as an agent to kill him/her and an important predictor of later suicide attempts and completions [2]. According to the World Health Organization (WHO) report, every year over 16,000,000 people attempt suicide, and 800,000 people die by suicide worldwide [3]. Worldwide, suicide ranked the 14[th] leading cause of mortality and morbidity. By the year 2030, it is expected to increase by 50%, becoming the 12th leading cause of death [4]. Globally, it is the major public health issue that is ranking the second cause of death for women ages between 15–29 [5], and the leading possible cause of death among pregnant mothers [6].

Pregnancy is mostly a sensitive time for women, being frequently both physically and mentally distressing. Currently, suicide is recognized as one of the major reasons for the death of women in middle-income and low-income countries [7]. Epidemiological evidence shows that SI among pregnant women is more common and higher than in the general population. Studies indicated that the estimated prevalence of SI among pregnant women ranges from 13.1% to 33% [8]. Antenatal SI has been often linked with an increased risk of subsequent suicidal attempts and death from suicide [6]. In Canada, during pregnancy 5% of women died due to suicide by using the means of committing suicide such as hanging and jumping from a high place [9].

In Africa, the estimated prevalence of SI among pregnant women ranged between 12–21% [10, 11]. Pregnancy helps women to have regular health services which is the golden chance for healthcare providers to intervene quickly if the prevalence of suicide is well understood [12]. However, efforts to study suicide in pregnancy have been hindered by social stigma and practical restrictions including inadequate data sources picking pregnancy and delivery status of patients with suicidal behavior [12].

SI among the pregnant population is associated with numerous consequences that adversely affected maternal and infant outcomes including fetal growth restriction, premature labor, cesarean delivery, respiratory distress, depression, and addicted alcohol [13–15].

There are several risk factors for maternal suicidal behavior during the perinatal period. Some of the risk factors that exacerbate suicidal ideation women who are being unmarried, lack of support, comorbid mental illnesses, have low educational attainment, unemployment, unplanned pregnancy, history of childhood abuse, intimate partner violence, and preexisting vulnerability such as, a family history of suicide, impulsivity, and previous and/or current psychiatric diagnoses including depression [8, 14, 16–18].

Despite this burden and consequences, in low and middle-income countries, there is a limited study on the prevalence of suicidality among pregnant women. To the best of our

knowledge, there are no studies in Ethiopia on the subject. Therefore, this study aimed to assess the prevalence of suicidal ideation and identify the associated factors among pregnant women to fill the existing gap in the literature.

# Methods and materials

## Study area, design, and period

An institution-based cross-sectional study was conducted in August 2020 at Jimma medical center, which is geographically located in the city of Jimma that is situated 352 km from Addis Ababa to the southwest part of Ethiopia. The center delivers service to the catchment population of about 15 million people. The ANC clinic provided services for a total of approximately 9850 women every year by many professionals including specialists, residents, general practitioners, midwives, and nurses [19].

## Source population

All pregnant women attending antenatal care at Jimma Medical Center.

## Study population

All pregnant women attending antenatal units were available during the study period.

## Inclusion and exclusion criteria

**Inclusion criteria.** All pregnant women age 18 and above, who had gestation age 30 days and above.

**Exclusion criteria.** Pregnant women who were critically ill and difficult to communicate.

## Sampling procedure and sampling techniques

**Sample size estimation.** A single population proportion formula was used to estimate the sample size. Sample size with z-value of 1.96 and marginal error of 5% sample was calculated as:-

$$n = \frac{(Z\,\alpha/2)^2\,P\,(1-P)}{d^2}$$

Where n = initial sample size a = confidence interval (95%) p = proportion of = 0.5
d = marginal error of 5% $(z\,\alpha/2)^2$ = 1.96

$$n = \frac{1.96^2 \text{ x } (0.5\,(1-0.5) = 384}{(0.05^2)}$$

Considering a 10% non-response rate a total sample of 423 pregnant women were included in the study.

**Sampling procedure.** A systematic sampling technique was used to recruit participants. The sampling interval was determined by dividing the total population who had follow up during a month of data collection period in OPDs of the JMC ANC unit by the sample size. Selection skip interval was, by taking total pregnant women of 848 (N) per and sample size (n) $423 = \frac{N}{n}, k = \frac{848}{423} = 2.01 = 2$, so the participants were selected every 2nd interval, the first woman was selected from the first two by lottery method who had to follow up during the data collection period.

## Data collection method and tools

A semi-structured questionnaire was used which has different subunits, questionnaires to assess socio-demographic factors, obstetrical factors, clinical factors, psychosocial factors, and substance use factors.

Symptoms of maternal depression were assessed using the Edinburgh Postnatal Depression Scale (EPDS) [20]. EPDS is a common tool for screening depressive symptomatology; initially, for use during the postnatal periods, it is also additionally validated for use during the perinatal periods in different countries and settings [21–23]. It was also validated among the perinatal population in Ethiopia [24]. It consists of 10 items questions that examine emotional state happening for at least the past 7 days. Each question score has four possible answers with an interval of 0–3. The maximum score is 30. To consider most seriously depressed women, similar to the previous study, if the score is 13 and above is used to recognize probable cases [20, 25]

Anxiety was assessed using the anxiety subscale adapted from the Depression, Anxiety, and Stress Scale (DASS -21). Each item contributes 0 to 3 points to the total score resulting in a total score that intervals from 0 to 21 and the score 8 and above was considered as having anxiety [26]. The IPV was assessed using the Abuse Assessment Scale (AAS). It is the most widely used tool to assess abuse among pregnant mothers in clinical settings. Women who gave responses yes to questions 2,3 or 4 were considered as having abuse [27].

The childhood physical and sexual abuse questionnaire was used to assess information regarding participants' experiences with physical and sexual abuse in childhood happening earlier than the age of 18 years [28].

Stress was assessed by the perceived stress scale (PSS). The PSS has 10 items multiple-choice self-report psychological instruments for measuring the perception of stress. Each item contributes 0 to 4 points to the total score resulting in a total score that intervals from 0 to 40, a higher score indicating greater perceived stress occurring one month before the interview [29]. In Ethiopia which also used to assess stress among pregnant women, teachers, and students [30–32].

Social support exploring family support, relationship with friends, partner/spouse help, conflict with spouse/partner, control feeling by spouse and, feeling unloved by spouse /partner during pregnancy, strengthening was assessed by the Maternity Social Support Scale (MSSS). It has three categories; low social support (less than 18), medium social support (18–23), and high social support (for scores 24–30) [33].

The Pittsburgh Sleep Quality Index (PSQI) was assessed sleep quality during pregnancy [34]. The PSQI contains 19 items which are categorized into seven components: subjective sleep quality, sleep latency, sleep duration, habitual sleep efficiency, sleep disturbances, use of sleeping medication, and daytime dysfunction. Each component scores ranging from 0 to 3 and then getting a global score with an interval from 0–21. A global score greater than 5 showed poor sleep quality and scores equal to or less than 5 are measured as good quality sleep. This yields sensitivity and a specificity of 89.6% and 86.5% respectively [34].

The suicidality module of the World Mental Health (WMH) survey initiative version of the World Health Organization (WHO) composite international diagnostic interview (CIDI) was used to assess suicidal ideation [35]. Which was also used to assess suicide among patients with Tuberculosis, Epilepsy, and HIV/AIDS [36–39]. In Ethiopia in which its Amharic version is validated Ethiopia both in clinical and community settings [40, 41].

Substance use was assessed by the WHO student drug-use questionnaire [42]. The presence of a known chronic medical such as diabetes mellitus, hypertension or others, family mental illness, family history of suicidal attempt was assessed by self-report (yes/ no response).

## Data collection procedure

Five BSc psychiatry professionals and one supervisor from the 1st year postgraduate student in psychiatry were trained on how to collect data. Each section of questionnaires was prepared in English and then translated into the local language Amharic and Afan Oromo, and back-translated to English by an independent person to ensure its understandability and consistency. A two days training of supervisor and data collectors was given on the purpose of the study, tools, how to collect data, sampling techniques, how to keep confidentiality, and how to handle ethical issues was discussed with the data collectors. The pre-test was conducted among 21 (5%) of the sample size pregnant women in Agaro General Hospital formerly the main study was done to recognize impending problems in the proposed study such as data collection tools and to check the performance of the data collectors. Regular supervision by the supervisor and principal investigator was made to ensure that all necessary data was appropriately collected. Each day throughout data collection completed questionnaires were checked for completeness and consistency. The collected data were edited and entered into the computer from a paper then checked twice and processed timely.

## Data processing and analysis

The data were entered into the Epi-Data version 3.1, and then data was exported to SPSS 25.0 version for cleaning and analysis. Descriptive statistics including frequencies, percentages, and summary statistics (mean values, and standard deviations) were calculated to define the study population about relevant variables. The bivariate logistic analysis was done to select candidate variables. All variables p-value < 0.25 in the bivariate analysis were entered into the multivariable logistic regression model. Multivariable logistic regression analysis was employed to control for possible confounding effects and to determine the presence of a statistically significant association between independent variables and outcome variables. The model of fitness was checked by Hosmer and Lemeshow goodness. A P-value < 0.05 was considered statistically significant and the strength of the association was presented by an odds ratio of 95% C.I.

## Ethics approval and consent to participate

Before the study begins ethical clearance was obtained from the ethical review committee of Jimma University. Then data collection was initiated after a letter of the corporation that is obtained from the above responsible office. Official permission was secured from JMC and the ANC unit coordinator. Written Informed consent was taken from each of the pregnant women and the information from individual mothers was kept confidential, their identity was not shown and there was no dissemination of the information without the respondent's permission. The data given by the participants was used only for research purposes. We prepared a private room for an interview; those women who reported suicidal thoughts or attempted and depression were immediately referred to mental health facilities (emergency) for further evaluation and management. Interviewers were trained to link participants found to be in physically risky conditions and/or in immediate need of counseling to psychologists and psychiatrists.

## Result

### Socio-demographic characteristics of participants

A total of 415 participants were included in the study, which resulted in an overall response rate of 98.1%. The mean age (± SD) of the respondents was 25.22(±4.62), with an age range of 18–38 years. Of all respondents, the majority were age range of 20–24 years 164(39.5%). About one-half (51.8%) of participants were Muslim religious followers. About three fourth of the

**Table 1. Socio-demographic characteristics of pregnant women attending antenatal care at Jimma medical center, Jimma, Southwest Ethiopia, 2020 (N = 415).**

| Variables | Categories | Frequency(n = 415) | Percent (%) |
|---|---|---|---|
| Age | 18–19 | 36 | 8.7 |
| | 20–24 | 164 | 39.5 |
| | 25–29 | 128 | 30.8 |
| | 30–34 | 73 | 17.6 |
| | > = 35 | 14 | 3.4 |
| Religion | Muslim | 215 | 51.8 |
| | Orthodox | 113 | 27.2 |
| | Protestant | 81 | 19.5 |
| | Others* | 6 | 1.5 |
| Marital status | Married | 313 | 75.4 |
| | Single | 74 | 17.8 |
| | Divorced | 23 | 5.5 |
| | Widowed | 5 | 1.3 |
| Ethnicity | Oromo | 251 | 60.5 |
| | Amhara | 69 | 16.6 |
| | Yeme | 35 | 8.4 |
| | Keffa | 29 | 7.0 |
| | Gurage | 20 | 4.8 |
| | Others** | 11 | 2.7 |
| Education status | Have no formal education | 32 | 7.7 |
| | Primary | 120 | 28.9 |
| | Secondary | 127 | 30.6 |
| | College and above | 136 | 32.8 |
| Occupational status | Government employed | 88 | 21.2 |
| | Merchant | 38 | 9.2 |
| | Farming | 14 | 3.4 |
| | Student | 67 | 16.1 |
| | Housewife | 163 | 39.3 |
| | Private employed | 45 | 10.8 |
| Residence | Urban | 320 | 77.1 |
| | Rural | 95 | 22.9 |
| Average monthly income (Eth. Birr) | <2166 | 268 | 64.6 |
| | > = 2166 | 147 | 35.4 |

Others

*Adventist &Catholic

**Tigre, Wolyita & Dawro

participants (75.4%) were married and 251(60.5%) were Oromo in their ethnicity. The educational status of participants showed that 136(32.8%) of them attended college and above. Regarding occupational status, 163(39.3%) were housewives. Large numbers of respondents were urban residents 320(77.1%). The majority 268 (64.6%) of study participants had an average monthly income below 2166 Ethiopian birr (Table 1).

## Obstetrics related characteristic of the participants

Nearly half of the study participants (47.0%) were in third-trimester pregnancy followed by first trimester 112 (27%) in their gestational age. Approximately two-thirds 269 (64.8%) and

**Table 2. Description of obstetrics-related factors among pregnant women attending antenatal care at Jimma medical center, Jimma, Southwest Ethiopia, 2020 (N = 415).**

| Variables | Categories | Frequency(n = 415) | Percent (%) |
|---|---|---|---|
| Pregnancy by trimester | First trimester | 112 | 27.0 |
| | Second trimester | 108 | 26.0 |
| | Third trimester | 195 | 47.0 |
| Gravidity | Primigravida | 146 | 35.2 |
| | Multigravida | 269 | 64.8 |
| Parity | Nullipara | 107 | 25.8 |
| | Multipara | 308 | 74.2 |
| History of abortion | Yes | 84 | 20.2 |
| | No | 331 | 79.8 |
| Abortion intention in the current pregnancy | Yes | 20 | 4.8 |
| | No | 395 | 95.2 |
| Current pregnancy status planned | No | 133 | 32 |
| | Yes | 282 | 68 |

three fourth 308 (74.2%) of the respondents were multigravida and multipara respectively. Out of the total participants, 84 (20.2%) women had a previous history of abortion, and 64 (15.4%) had abortion intentions in the current pregnancy. More than two-thirds of 282(68.0%) of the women had a planned pregnancy (Table 2).

## Clinical and substance-related factors of the participants

According to this study finding, 21(5.1%) of respondents had a history of mental illness. Among participants, 43(10.4%) of respondents had a family history of mental illness and 40 (9.6%) participants reported a family history of suicidal attempts. From respondents, 23(5.5%) women had a comorbid medical illness, from these medical illnesses, HIV/AIDS 4(1%), asthma 5(1.2%), diabetes 6(1.4%), and hypertension 8(1.9%) were reported. Of the participants, 114(27.5%) and 141(34.0%) had depression and anxiety symptoms respectively. Regarding sleep quality, (30.8%) of the respondents had reported poor sleep quality (Table 3).

Regarding the current use of the substance, 35(8.4%) of the respondents had a history of substance use within the past three months before data collection time. Among users majority of them, 19(4.6%) used alcohol, 13(3.1%) of the respondents were chewing khat and 3(0.7%) were smoking a cigarette within the past three months (Fig 1).

## Psychosocial factors of the participants

From the total of the participants, 94(22.7%) of the respondents had experienced recent violence from their intimate partner and 65(15.7%) of the women had reported a history of childhood abuse. About one-third (34.7) of the respondents had stress during pregnancy. Regarding social support, more than half (53.3%), 141(34.0%), and 53 (12.8%) of the pregnant women had received medium social support, high social support, and poor social support respectively (Table 4).

**The magnitude of suicidal ideation among pregnant women attending antenatal care at Jimma, Ethiopia, 2020(n = 415).** In the present study, the prevalence of suicidal ideation among pregnant women was 13.3% (95%CI:10.1–16.4) (Table 5).

## Factors associated with suicidal ideation among pregnant women

In the bivariate analysis, marital status, income, parity, educational status, history of abortion, unplanned pregnancy, family history of mental illness, family history of suicidal attempt,

**Table 3. Description of clinical related factors among pregnant women attending antenatal care at Jimma medical center, Jimma, Southwest, Ethiopia, 2020 (n = 415).**

| Variables | Categories | Frequency(n = 415) | Percent (%) |
|---|---|---|---|
| Past mental illness history | Yes | 21 | 5.1 |
| | No | 394 | 94.9 |
| Family history of mental illness | Yes | 43 | 10.4 |
| | No | 372 | 89.6 |
| Family history of suicidal attempt | Yes | 40 | 9.6 |
| | No | 375 | 90.4 |
| Chronic medical illness | Yes | 23 | 5.5 |
| | No | 392 | 94.5 |
| HIV/AIDS | Yes | 4 | 1.0 |
| | No | 411 | 99.0 |
| Asthma | Yes | 5 | 1.2 |
| | No | 410 | 98.8 |
| Diabetes | Yes | 6 | 1.4 |
| | No | 409 | 98.6 |
| Hypertension | Yes | 8 | 1.9 |
| | No | 407 | 98.1 |
| Depression | Yes | 114 | 27.5 |
| | No | 301 | 72.5 |
| Anxiety | Yes | 141 | 34.0 |
| | No | 274 | 66.0 |
| Sleep quality | Poor | 128 | 30.8 |
| | Good | 287 | 69.2 |

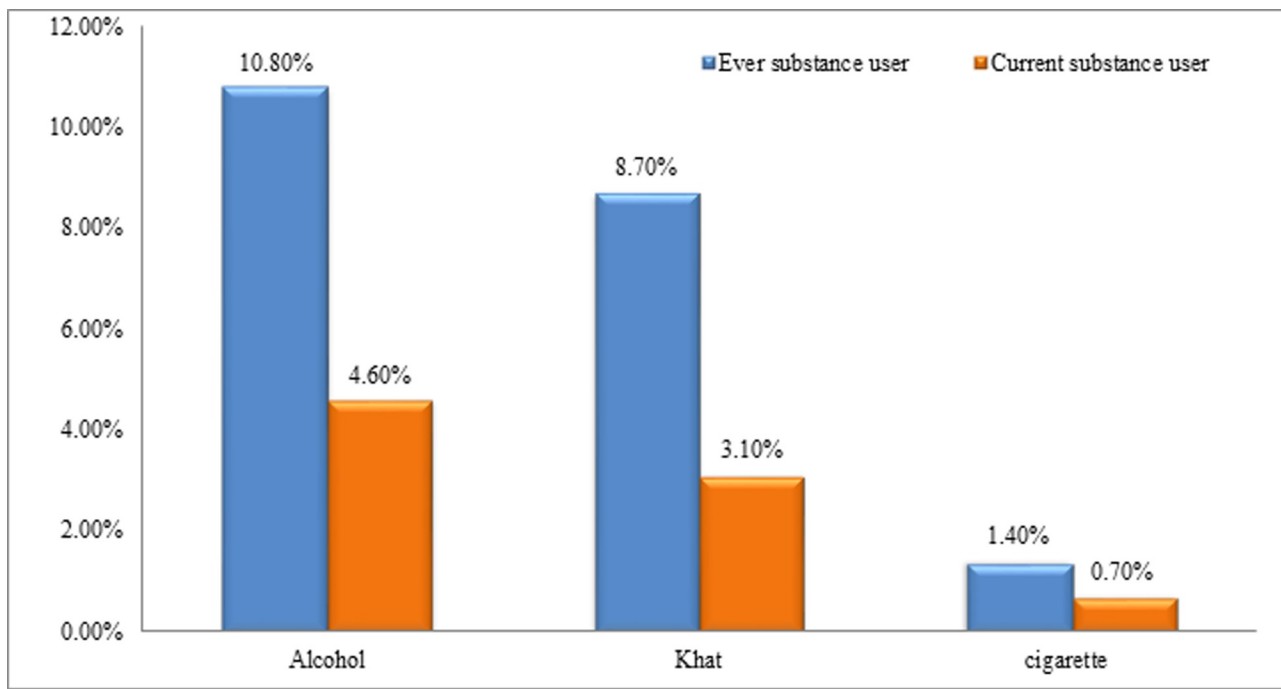

**Fig 1. Ever and current substance use among pregnant women attending antenatal care at Jimma medical center, Jimma, Southwest, Ethiopia, 2020 (n = 415).**

**Table 4. Psychosocial factors among pregnant women attending antenatal care at Jimma medical center, Jimma, Southwest, Ethiopia, 2020 (n = 415).**

| Variables | Categories | Frequency(n = 415) | Percent (%) |
|---|---|---|---|
| Intimate partner violence | Yes | 94 | 22.7 |
| | No | 321 | 77.3 |
| Childhood abuse | Yes | 65 | 15.7 |
| | No | 350 | 84.3 |
| Stress | Yes | 144 | 34.7 |
| | No | 271 | 65.3 |
| Social support | Low social support | 53 | 12.8 |
| | Medium social support | 221 | 53.2 |
| | High social support | 141 | 34.0 |

depression, anxiety, poor sleep quality, stress, intimate partner violence, and social support showed a p-value of <0.25 and became a candidate for multivariable analysis. In multivariable binary logistic regression variables; marital status, depression, anxiety, history of abortion, sleep quality, stress, and intimate partner violence were found to be statistically associated with suicidal ideation at a p-value less than 0.05.

The odds of suicidal ideation among participants with the marital status category of (single, widowed, divorced) was 2.8 times higher as compared to married women [AOR = 2.80;95% CI (1.23,6.37)]. Those pregnant women who had a previous history of abortion were 2.45 times

**Table 5. Distribution of suicidal ideation and attempt among pregnant women attending antenatal care at Jimma medical center, Jimma, Southwest, Ethiopia, 2020 (N = 415).**

| Variables | Categories | Frequency(n = 415) | Percent (%) |
|---|---|---|---|
| Ever suicidal ideation | Yes | 78 | 18.8 |
| | No | 337 | 81.2 |
| Suicidal ideation in 1 month | Yes | 55 | 13.3 |
| | No | 360 | 86.7 |
| Ever plan of suicide | Yes | 31 | 7.5 |
| | No | 384 | 92.5 |
| Ever suicide attempt | Yes | 30 | 7.2 |
| | No | 385 | 92.8 |
| Suicidal attempt in 1 month | Yes | 12 | 2.9 |
| | No | 403 | 97.1 |
| Frequency of suicide in 1 month | Once | 8 | 66.7 |
| | Twice | 3 | 25 |
| | More than twice | 1 | 8.3 |
| Reason for 1-month suicidal attempt | Family conflict | 5 | 41.7 |
| | Death in family | 2 | 16.7 |
| | Financial constraint | 3 | 25 |
| | Relationship problems | 2 | 16.6 |
| Severity related to 1-month attempt | Seriously attempted | 6 | 50 |
| | Ineffective method | 4 | 33.3 |
| | To seek help | 2 | 16.7 |
| Methods of 1-month attempt | Poisoning | 8 | 66.7 |
| | Hanging | 3 | 25 |
| | Sharp tools | 1 | 8.3 |

more likely to have suicidal ideation as compared with respondents who did not have a history of abortion [AOR = 2.45;95%CI(1.03,5.93)].

Those women with depression were about 4 times more likely to have suicidal ideation than their counterparts [AOR = 4.28;95%CI (1.75,10.44)]. Regarding anxiety, the participants with anxiety were about 3 times more likely to have suicidal ideation than their counterparts [AOR = 2.99; 95%CI (1.24,7.20)]. Likewise, participants with poor subjective sleep quality were 2.85 times more likely to have suicidal ideation as compared with women who had good sleep quality [AOR = 2.85; 95%CI (1.19, 6.79)].

Furthermore, the odds of having suicidal ideation among women who had stress was about 2.50 times higher as compared with the referent groups [AOR = 2.50;95%CI(1.01,5.67)]. Finally, pregnant mothers who reported violence from an intimate partner were 2.43 times more likely to have suicidal ideation as compared with those who did not experience Intimate partner violence [AOR = 2.43; 95%CI (1.07, 5.47)] (Table 6).

## Discussion

To the best of our knowledge, this study is the first to determine the prevalence and associated factors of suicidal ideation among pregnant women in Ethiopia. The result indicated that the magnitude of suicidal ideation during the current pregnancy was 13.3% [(95% CI, 10.1–16.4)]. This result was comparable with other findings done in Brazil 10.3% [43], Pakistani 11% [44], South Africa (12%) [10], and Southern Brazil 13.3% [45]. On the other hand, this study finding was higher when compared with a study done in Brazil 8.1% [46], Peru 8.8% [47], Brazil 6.3% [48], the USA 4.6% [49], and India 7.6% [50]. The discrepancy might be due to the study setting and inclusion criteria. For example, in India, a study was conducted on urban setting pregnant women only who were between 5 and 20 weeks of pregnancy while the current study was done on rural and urban and pregnant women at all trimesters of gestation [50]. The difference in assessment tool might be another possible reason, suicidal ideation assessed using the 10th item of the Edinburgh Postnatal Depression Scale (EPDS) which evaluated for a week of duration only in Brazil [46]. Another discrepancy might be due to the difference in study design, study setting, sample size, social support practice, and sampling technique, the socio-demographic and cultural context of the women.

However, in some other studies, the proportion of suicidal ideation was higher than the current study, a study conducted in Brazil 23.53% [51], Egypt 20.4% [11], and Peru 16.8% [52]. This variation might be due to the screening tool difference in which a previous study MINI was used in Brazil [51] and Beck Scale for Suicide Ideation (BSS) in Egypt [11], whereas in this study CIDI was used [35]. Also, sample size difference might be another possible reason for their incongruence in Peru which was 641 study participants included [52]. Another possible reason might be the difference in participants who had different socio-economic and demographic characteristics in the populations.

Regarding the associated factors, in this study, marital status categories with a lack of cohabiting partners (single, divorced, and widowed) were nearly three times more likely to have suicidal ideation during the pregnancy period as compared to married women. This finding was in agreement with different studies in the USA [8, 18] and Brazil [46, 48, 53]. The reason might be due to the lower level of perceived social and emotional support from families and intimate partners. Another possible reason might be due to the socio-cultural value in the communities, some communities give high value to married ones. Moreover, this condition can directly affect women's mental health particularly during pregnancy [10].

In the current study, we found that women who had depression were 4 times more likely to have suicidal ideation than undepressed participants. Similar to a finding of different studies

**Table 6. Bivariate and multivariate logistic regression analysis results of suicidal ideation among pregnant women attending ANC at JMC, Jimma, Southwest Ethiopia, 2020 (N = 415).**

| Variables | Category | Suicidal ideation | | COR(95%C.I) | AOR(95%C.I) | P-values |
|---|---|---|---|---|---|---|
| | | Yes (%) | No (%) | | | |
| Marital status | Lack of cohabit partner | 33(32.4%) | 69(67.6%) | 6.33(3.47,11.53) | 2.80(1.23,6.37) | **0.014***|
| | Married | 22(7.0%) | 291(93%) | 1 | 1 | |
| Income (Ethio birr) | <2166 | 40(14.9%) | 228(85.1%) | 1.54(0.82. 2.90) | 0.52(0.21,1.30) | 0.162 |
| | > = 2166 | 15(10.2%) | 132(89.8%) | 1 | | |
| Educational status | No formal education | 4(12.5%) | 28(87.5%) | 1.35(0.41,4.45) | 0.63(0.125,3.20) | 0.58 |
| | Primary | 14(11.7%) | 106(88.3%) | 1.25(0.56, 2.77) | 0.75(0.24,2.36) | 0.62 |
| | Secondary | 24(18.9%) | 103(81.1%) | 2.20(1.07, 4.54) | 1.74(0.60,5.02) | 0.31 |
| | College and above | 13(9.6%) | 123(90.4%) | 1 | 1 | |
| Parity | Null | 28(26.2%) | 79(73.8%) | 3.69(2.07,6.62) | 1 | 0.052 |
| | One or more | 27(8.8%) | 281(91.2%) | 1 | 2.26(0.99,5.16) | |
| History of abortion | Yes | 27(32.1%) | 57(67.9%) | 5.13(2.81, 9.34) | 2.45(1.03,5.93) | **0.042***|
| | No | 28(8.5%) | 303(91.5%) | 1 | | |
| Current pregnancy planned | No | 29(21.8%) | 104(78.2%) | 2.74(1.54, 4.90) | 1 | 0.12 |
| | Yes | 26(9.2%) | 256(90.8%) | 1 | 1.91(0.84,4.32) | |
| Family history of mental illness | Yes | 13(30.2%) | 30 (69.8%) | 3.41(1.65,7.03) | 0.98(0.29,3.25) | 0.97 |
| | No | 42(11.3%) | 330(88.7) | 1 | | |
| Family history of suicidal attempt | Yes | 9(22.5%) | 31(77.5%) | 2.07(0.93,4.64) | 0.37(0.11,1.31) | 0.12 |
| | No | 46(12.3%) | 329(87.7%) | 1 | | |
| Depression | Yes | 45(39.5%) | 69(60.5%) | 18.98(9.11,39.53) | 4.28(1.75,10.44) | **0.001***|
| | No | 10(3.3%) | 291(96.7%) | 1 | | |
| Anxiety | Yes | 43(30.5%) | 98(69.5%) | 9.58(4.85,18.92) | 2.99(1.24,7.20) | **0.015***|
| | No | 12(4.4%) | 262(95.6%) | 1 | | |
| Sleep quality | Poor | 42(32.8%) | 86(67.2%) | 10.29(5.28,20.06) | 2.85(1.19,6.79) | **0.018***|
| | Good | 13(4.5%) | 274(95.5%) | 1 | | |
| Stress | Yes | 36(25.0%) | 108(75.0%) | 4.42(2.43, 8.05) | 2.50(1.01,5.67) | **0.03***|
| | No | 19(7.0%) | 252(93.0%) | 1 | | |
| Intimate partner violence | Yes | 35 (37.2%) | 59(62.8%) | 8.93(4.82, 16.53) | 2.43(1.07,5.47) | **0.033***|
| | No | 20(6.2%) | 301(93.8%) | 1 | | |
| Social support | Low social support | 19(35.8%) | 34(64.2%) | 7.32(3.11,17.19) | 1.09 (0.56,2.11) | 0.80 |
| | Medium social support | 26(11.8%) | 195(88.2%) | 1.75(0.815,3.74) | 1.08(0.56,2.12) | 0.81 |
| | High social support | 10(7.1%) | 131(92.9%) | 1 | 1 | |

*Statistically significant at P-value < 0.05, AOR, Adjusted odds Ratio, 1 = reference category, Chi square = 8.8, Hosmer Lemeshow goodness-of-fit 0.52, degrees of freedom = 7, Maximum VIF = 1.49.

from the USA [8, 14, 18, 52], two studies from Brazil [46, 53], India [50] as well as in Egypt [11] support this finding; we also observed a strong association between antenatal depressive symptoms and suicidal ideation. The possible reason might be the presence of hormonal changes during pregnancy could be a risk for depression. Besides that depression decreases the level of the neurotransmitter serotonin, in which studies had shown an association between decreased level of serotonin and suicidal behavior [54]. It may also be due to the direct effect of depression which makes individuals feel hopeless, isolated, and worthless.

This finding also revealed that pregnant women with anxiety were three times more likely to have suicidal ideation than their counterparts. This was supported by the study conducted in the USA [8], Pakistani [44], South Africa [10], and Egypt [11]. Pregnancy is a more sensitive

period for women and becomes distressing either physically or mentally. Pregnancy and previous experience of birth can situate women into conditions outside their comfort zones, which also cause anxiety; anxious concern will be about the health of their baby, fear of experiencing give birth or worry about weight gain, body shape, and being a responsible parent. Therefore the anxiety might be a potential cause that leads to suicidal behaviors [55].

The current study also showed that women who had experienced intimate partner violence were 2.43 times more likely to have suicidal ideation as compared to those women who didn't experience intimate partner violence. The finding was consistent with a study conducted in the U.S [18], Brazil [45], India [50], Pakistani [44], and South Africa [10]. These studies support the association between IPV and suicidal ideation. This might be because IPV creates insecure relationships with their intimates and which also contributes to the development of SI [10]. Nevertheless, one study from the USA contradicts our finding, in which IPV was not significantly associated with suicidal ideation. The possible reason could be, in this study 69 (3.2%) of women experienced IPV from a total of 2159 participants, besides in the US, abusers made seriously asked and penalized by law. Whereas, in our study 415 samples were included; 94 (22.7%) experienced intimate partner violence [14].

Another predictor for antenatal suicidal ideation was the history of abortion those pregnant women who had a previous history of abortion were 2.45 times more likely to have suicidal ideation as compared with respondents who do not have a history of abortion. A current study finding was congruent with a finding from Southern Brazil [45]. This might be due to experiencing abortion could result in stressful situations that impose them to suicidal thoughts.

We also found that women who had poor subjective sleep quality were about three times more likely to have suicidal ideation as compared with those women who had good sleep quality. This was compatible with a study report from Brazil [53] and two studies from Peru [52, 56] that showed that poor sleep quality was significantly associated with women who had suicidal ideation. However, during pregnancy, there is a hormonal change, which usually potentially causes sleep alteration. Scholars also assessed sleep quality and its relationship with suicidal ideation during pregnancy; those who didn't get well sleep were more likely to have suicidal thoughts [52].

The odds of having suicidal ideation among women who had stress was about 2.50 times higher when compared with the referent groups. This was congruent with the study done in the USA that showed that perceived stress had been strongly associated with prenatal suicidal ideation [14]. Women who had experienced stress during pregnancy lead to changes in their mood, feeling of loss of control, and being frightened about the future. During this time of fear, women considered suicide as a way to escape from stressors [57].

The limitation of the study might be emanated from the tools used are not culturally validated. This might bring a difference in the study findings.

## Conclusion

Our study found that the prevalence of suicidal ideation and attempt among pregnant women was high. Lack of cohabiting partners, depression, anxiety, poor sleep quality, history of abortion, intimate partner violence, and stress were variables which are independent predictors of suicidal ideation among pregnant women. Screening and interventions of antenatal SI are warranted.

## Supporting information

**S1 File. Minimal data set.**
(XLSX)

## Acknowledgments

We would like to express our sincere thanks to Wollo University, Jimma University, Jimma Medical Center, study participants, data collectors, and supervisors. I would like to say thank you, Alemayehu Molla.

## Author Contributions

**Conceptualization:** Tamrat Anbesaw, Alemayehu Negash, Getinet Ayano.

**Data curation:** Tamrat Anbesaw, Alemayehu Negash, Almaz Mamaru, Asmare Belete.

**Formal analysis:** Tamrat Anbesaw, Alemayehu Negash, Asmare Belete, Getinet Ayano.

**Funding acquisition:** Habtamu Abebe.

**Investigation:** Tamrat Anbesaw.

**Methodology:** Tamrat Anbesaw, Alemayehu Negash, Almaz Mamaru, Habtamu Abebe, Asmare Belete, Getinet Ayano.

**Resources:** Tamrat Anbesaw.

**Software:** Tamrat Anbesaw, Almaz Mamaru, Habtamu Abebe, Asmare Belete, Getinet Ayano.

**Supervision:** Tamrat Anbesaw, Alemayehu Negash.

**Visualization:** Tamrat Anbesaw, Alemayehu Negash.

**Writing – original draft:** Tamrat Anbesaw, Almaz Mamaru, Habtamu Abebe, Asmare Belete, Getinet Ayano.

**Writing – review & editing:** Tamrat Anbesaw, Alemayehu Negash, Almaz Mamaru, Asmare Belete.

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
