## [Decision Letter · Decision Letter 0]

14 May 2021

PONE-D-21-09284

SUICIDAL IDEATION AND ASSOCIATED FACTORS AMONG PREGNANT WOMEN ATTENDING ANTENATAL CARE AT JIMMA MEDICAL CENTER, JIMMA, SOUTHWEST ETHIOPIA, 2020.

PLOS ONE

Dear Dr. Anbesaw,

Thank you for submitting your manuscript to PLOS ONE. After careful consideration, we feel that it has merit but does not fully meet PLOS ONE’s publication criteria as it currently stands. Therefore, we invite you to submit a revised version of the manuscript that addresses the points raised during the review process.

We look forward to receiving your revised manuscript.

Kind regards,

Russell Kabir, PhD

Academic Editor

PLOS ONE

Journal Requirements:

5. Please amend your list of authors on the manuscript to ensure that each author is linked to an affiliation. Authors’ affiliations should reflect the institution where the work was done (if authors moved subsequently, you can also list the new affiliation stating “current affiliation:….” as necessary).

7. Please ensure that you refer to Figure 1 in your text as, if accepted, production will need this reference to link the reader to the figure.

8. We note you have included a table to which you do not refer in the text of your manuscript. Please ensure that you refer to Table 4 in your text; if accepted, production will need this reference to link the reader to the Table.

Reviewers' comments:

Reviewer's Responses to Questions

**Comments to the Author**

1. Is the manuscript technically sound, and do the data support the conclusions?

Reviewer #1: Yes

Reviewer #2: Yes

Reviewer #3: Partly

2. Has the statistical analysis been performed appropriately and rigorously? 

Reviewer #1: Yes

Reviewer #2: Yes

Reviewer #3: Yes

3. Have the authors made all data underlying the findings in their manuscript fully available?

Reviewer #1: Yes

Reviewer #2: Yes

Reviewer #3: Yes

4. Is the manuscript presented in an intelligible fashion and written in standard English?

Reviewer #1: Yes

Reviewer #2: Yes

Reviewer #3: No

5. Review Comments to the Author

Reviewer #1: Important investigation. Thanks. I have some suggestions to improve the paper.

Title: Could be more precise. To me some unnecessary things are there.

Abstarct_Method: Please mention the name of all the instruments used.

Conclusion: Seems repetition of the results

Introduction: Suicidal ideation (SI) is a passive thought... this is not correct. It can be both active and passive. Please check the context of the reference and reconfirm

Rationale is grossly absent. The study was done in Ethiopia not in all LAMICS. I was wondering why it has been doen in Ethiopia with substantial contextualizTION.

Methods: August 01-30/ 2020 at Jimma,,, August 2020 is enough

BSc psychiatry professionals?? could you please explain.

Results: Table 3: comorbid depression.... Please make it uniform i.e. if depression is comorbid why not anxiety and other disorder? I think mentioning depression indcate that it is comorbid

Discussion: I was looking for the limitations section while it should be mentioned that all intruments are not culturally validated.

Conclusion: Seems repetition of results.

Reviewer #2: a. The authors investigated an important area “Suicidal behaviour among pregnant women”. It is an area that is poorly studied. The present study found a significantly higher prevalence of suicidal ideation among Ethiopian pregnant women. This crucial information will be useful for the policy makers to augment the existing suicide prevention program.

b. How the authors have selected the catchment area for selection of participants?

c. Whether the authors have used any screening tool for assessment of Psychiatric disorders? Psychiatric disorder is an important attributing factor for suicidal behavior. It is important to look for it.

d. Refer to page 16: AT many places it has been mentioned (Error! Reference source not found.). Kindly provide the reference.

Reviewer #3: The author appears to assess the “prevalence and risk factor of suicidal ideation among pregnant women attending antenatal care at a medical center in Southwest Ethiopia” but the title of this manuscript says “SUICIDAL IDEATION AND ASSOCIATED FACTORS AMONG PREGNANT WOMEN ATTENDING ANTENATAL CARE AT JIMMA MEDICAL CENTER, JIMMA, SOUTHWEST ETHIOPIA, 2020”. I suggest this title needs to be modified to this: “Prevalence and risk factors of suicidal ideation in Ethiopia: A case study of pregnant women attending antenatal care at Jimma medical center in Southwest Ethiopia”.

Please find below my review details:

Abstract:

Under background, the author will need to recast the aim of the study as the year is not necessary.

Line 1 in methods, the author conducted the study with 415 pregnant women. This is not the same number in the methodology section (423 mentioned). Also, the year of the study is not mentioned at all here and this is where it is really necessary.

Line 1 in result, the author mentioned “current pregnancy”, he may need to recast the entire sentence.

Background:

Paragraph 4 line 2, the author made some tautologies, like “including such as psychiatric disorders”. This needs to be looked into and corrected.

Author needs to recast the whole of Paragraph 5. The message that is being communicated is not clear.

In paragraph 6, the author mentioned that there “there are no studies in Ethiopia on the subject”. I conducted a quick search and found Amare et al., 2018; Bifftu et al., 2019 and Leul et al., 2021. With this, it shows clearly that the author has not done enough literature search and needs to do more.

Methods and Materials:

Under study area, design and period, the author did not cite any reference.

Author also touched on study area only, no information was provided on the study design and period or should one say that this entire section had not been well itemized. The author did not make clear if Source population, Study population, Inclusion and exclusion criteria and sampling procedure techniques are all under design and period as they are just listed here.

The sample size here is 423 as against what is in the abstract.

Author made a statement “similar to the previous study” in line 8 paragraph 2 under Data collection and method tools. I thought the author already said this is the first study?

The author may need to recast the entire Data collection procedure section. It may not be necessary to include the degree of the psychiatric professionals and stating that the supervisor is a 1st year postgraduate student may not be necessary.

Results:

In this section, author needs to do a complete overhaul especially with typographical errors.

Under obstetrics related characteristics of the participants, the author should delete “In the current study”, this is not needed.

Line 3 and 4, Paragraph 2 under clinical and substance-related factors of the participants, the author gave percentages that are not correct. The percentages are supposed to be subset of 35 and not of the total. Therefore, the correct percentages should be 19 (54.2%), 13(37.1%) and against what it is in the manuscript.

The author should remove all the error messages also “reference source not found”.

Result presented under magnitude of suicidal ideation among pregnant women “where” are not well described. Author will need to touch some of the other data provided in the table.

Paragraph 1 under “Factors associated with suicidal ideation among pregnant women” needs to either be completely removed or recast. It looks more to me like materials and method than results.

Discussion:

This work is not well discussed. Author needs to conduct more literature search and discuss appropriately.  

Conclusion:

Author needs to recast and conduct more literature search to arrive at a logical conclusion

Ethics approval and consent to participate:

Author did not provide evidence of ethical approval for this study.

6. PLOS authors have the option to publish the peer review history of their article (what does this mean?). If published, this will include your full peer review and any attached files.

Reviewer #1: No

Reviewer #2: No

Reviewer #3: No

---

## [Author Response · Author response to Decision Letter 0]

9 Jul 2021

We are appreciating and thanking such like supporting comments and suggestions, it gives strength to do more. PLEASE ACCEPT OUR REVISED MANUSCRIPT. THANK YOU!

---

## [Editor Report · Decision Letter 1]

23 Jul 2021

SUICIDAL IDEATION AND ASSOCIATED FACTORS AMONG PREGNANT WOMEN ATTENDING ANTENATAL CARE IN JIMMA ,ETHIOPIA, 2020.

PONE-D-21-09284R1

Dear Dr. Tamrat,

We’re pleased to inform you that your manuscript has been judged scientifically suitable for publication and will be formally accepted for publication once it meets all outstanding technical requirements.

Kind regards,

Russell Kabir, PhD

Academic Editor

PLOS ONE
---

## [Editor Report · Acceptance letter]

11 Aug 2021

PONE-D-21-09284R1 

Suicidal Ideation nd Associated Factors among Pregnant Women attending Antenatal Care in Jimma Medical Center, Ethiopia. 

Dear Dr. Anbesaw:

I'm pleased to inform you that your manuscript has been deemed suitable for publication in PLOS ONE. Congratulations! Your manuscript is now with our production department. 

Kind regards, 

on behalf of

Dr. Russell Kabir 

Academic Editor

PLOS ONE